# Anion Exchange Membranes Prepared from Quaternized Polyepichlorohydrin Cross-Linked with 1-(3-aminopropyl)imidazole Grafted Poly(arylene ether ketone) for Enhancement of Toughness and Conductivity

**DOI:** 10.3390/membranes10070138

**Published:** 2020-06-30

**Authors:** Cao Manh Tuan, Vo Dinh Cong Tinh, Dukjoon Kim

**Affiliations:** School of Chemical Engineering, Sungkyunkwan University, Suwon, Gyeonggi 440-746, Korea; caomanhtuan26@gmail.com (C.M.T.); vodinhcongtinh@gmail.com (V.D.C.T.)

**Keywords:** Anion exchange membrane, ion conductivity, imidazole, crosslinking, PAEK

## Abstract

A novel anion exchange membrane was synthesized via crosslinking of the quaternized polyepichlorohydrin (QPECH) by 1-(3-aminopropyl) imidazole grafted poly(arylene ether ketone) (PAEK-API). While the QPECH provided an excellent ion conductive property, the rigid rod-structured PAEK-API played a reinforcing role, along with providing the high conductivity associated with the pendant API group. The chemical structure of QPECH/PAEK-API membranes was identified by ^1^H nuclear magnetic resonace spectroscopy. A variety of membrane properties, such as anion conductivity, water uptake, length swelling percentage, and thermal, mechanical and chemical stability, were investigated. The QPECH/PAEK-API1 membrane showed quite high hydroxide ion conductivity, from 0.022 S cm^−1^ (30 °C) to 0.033 S cm^−1^ (80 °C), and excellent mechanical strength, associated with the low water uptake of less than 40%, even at 80 °C. Such high conductivity at relatively low water uptake is attributed to the concentrated cationic groups, in a cross-linked structure, facilitating feasible ion transport. Further, the QPECH/PAEK-API membranes showed thermal stability up to 250 °C, and chemical stability for 30 days in a 4 NaOH solution, without significant loss of ion exchange capacity.

## 1. Introduction

Fuel cells (FC) are applied in many industrial fields, such as portable devices, vehicles and distributed power generation systems, as a potential eco-friendly power source [1,2,3,4]. In the development of polymer electrolyte membranes, anion exchange membranes (AEM) have recently drawn lots of attention because they have many advantageous characteristics over proton exchange membranes (PEM), as well as liquid electrolyte. For example, AEMs significantly reduce the carbonation and leakage problems of liquid electrolyte [5,6]; AEMs reduce fuel crossover more effectively than PEMs, because of their anion transfer direction opposite to the fuel [7,8,9]; AEMFC can accommodate much cheaper catalysts, such as cobalt, silver and nickel, than PEMFC, where platinum is usually used [10,11]. Furthermore, AEMFC exhibits fast fuel oxidation reactions in a high pH medium [12,13,14,15,16,17,18,19].

There has recently been synthesis of a variety of anion exchange membranes [2], based on polyetherketone, quaternized poly(aryl ether oxadiazole), poly(aryl ether oxadiazole) and others [16,17,18,20,21,22]. Although many attractive properties of the membrane were reported, some weaknesses were also pointed out, including low polymerization reaction yield, low anion conductivity, and low mechanical strength [23,24]. Recently, great attention has been paid to composite membrane systems containing reinforcing materials to improve the tensile strength of the polymer electrolytes [25,26]. These composite membranes, however, still often show a weakness associated with incompatibility between the polymer matrix and the supporting materials.

Among a variety of anion exchange groups [6], the quaternary ammonium is a well-established group typically applied in numerous AEMs. When the quaternary ammonium group is introduced to the polymer electrolyte membrane, many toxic and volatile reactants, such as trimethylamine and triethylamine, are occasionally released during the chloromethylation or quaternization process. Further, as aliphatic quaternary ammonium groups often cause the degradation of polymer chains via Hoffmann elimination and SN_2_ substitution reactions, this can result in the serious mechanical failure of the membranes after long-term utilization.

Polyepichlorohydrin (PECH) is a commercially available, cost competitive, high molecular weight polymer material. The highly concentrated pendant chloromethyl groups in PECH are quite attractive for their ability to establish high anion conductivity by a simple quaternization reaction, without external chloromethylation. When the pristine quaternized PECH (QPECH) is prepared as a membrane, however, it illustrates too-weak mechanical strength in a water environment, and thus it cannot sustain its role as a membrane throughout long-term operation. On the other hand, 1-(3-aminopropyl) imidazole grafted poly(arylene ether ketone) (PAEK-API) has shown good thermal, mechanical and chemical stability under fuel cell operation conditions in the previous works [21,22]. The critical requirement of PAEK-API membrane is the improvement of its conductivity. As its molecular structure contains carboxyl acid in pendant sites, it can be chemically cross-linked with chloromethyl groups of PECH, and thus the final crosslinked membrane structure may provide not only good thermal, mechanical and chemical stability, but also high anion conductivity. In this study, we used API functional groups, rather than well-known quaternary ammonium groups, for the ion conductive group, as API is expected to show higher anion conductivity and chemical stability in a high pH environment than quaternary ammonium groups, because of the π-conjugated structure of the imidazolium ring.

## 2. Experiment

### 2.1. Materials

Polyepicholorohydrin (PECH), 4,4-bis(4-hydroxyphenyl)-valeric acid, potassium carbonate (K_2_CO_3_), N,N′-dicyclohexylcarbodiimide (DCC), N-hydroxysuccinimide (NHS) and methyl iodide (CH_3_I) were purchased from Aldrich Chemical Company (Milwaukee, WI, USA). 4,4′-Difluorobenzophenone, dimethylsulfoxide (DMSO), N,N-dimethylacetamide (DMAc), tetrahydrofuran (THF), 1-(3-aminopropyl)imidazole (API) and 1-methylimidazole were purchased from Tokyo Chemical Industry (Japan). Toluene, isopropanol and methanol were purchased from Samchun Chemical Company (Korea), and hydrochloric acid was from Duksan Chemical Company (Korea).

### 2.2. Preparation of Anion Exchange Membrane

#### 2.2.1. Synthesis of PAEK-API Precursors

Poly(arylene ether ketone) with pendant -COOH group (PAEK-COOH) was firstly synthesized from 4,4-bis(4-hydroxyphenyl)-valeric acid and 4,4′-difluorobenzophenone in a DMSO and toluene mixture, in the presence of K_2_CO_3_ [27]. Biproducts were removed from the reaction between 4,4-bis(4-hydroxyphenyl)-valeric acid (0.01 mol) and K_2_CO_3_ (0.025 mol) at 145 °C for 4 h, followed by the step-growth polycondensation reaction with 4,4′-difluorobenzophenone (0.01 mol) at 150 °C for 12 h, and then at 168 °C for 24 h. The synthesized polymer was dissolved in HCl and THF mixture before precipitation in isopropanol. The final product, PAEK-COOH, was washed several times with isopropanol and distilled (DI) water, and then dried in a vacuum oven at 60 °C for 24 h.

PAEK-NHS was an intermediate precursor produced from PAEK–COOH. NHS (3.879 mmol) and DCC (3.879 mmol) were added into PAEK-COOH (3.23 mmol) solution in DMF (15 mL) at 1.2:1 molar ratio. The NHS substitution reaction was conducted under continuous stirring at room temperature for 12 h, and then at 40 °C for 12 h. After the PAEK-NHS solid product was obtained by precipitation in isopropanol, it was consecutively washed with isopropanol and methanol several times, before drying in a vacuum oven at 40 °C for 24 h. To prepare API grafted PAEK (PAEK-API), PAEK-NHS (0.01 mol) was dissolved in 20 mL DMAc under stirring for 5 h. After API (0.012 mol) was added to the PAEK-NHS solution in the molar ratio of 1.2:1, the solution was stirred for 3 h. The resulting solution was dropped into IPA for precipitation. The PAEK-API product was washed three times with isopropanol and then dried in an oven at 60 °C overnight. The synthetic scheme of PAEK-API is shown in Scheme 1.

#### 2.2.2. Preparation of QPECH/PAEK-API Membranes

QPECH/PAEK-API was synthesized by formation of crosslinks between the API groups of PAEK and the chloromethyl groups of PECH. Firstly, PAEK-API and PECH were separately dissolved in 10 mL DMSO, in the different molar ratios of PECH/PAEK-API of 5.8, 3.8, and 2.9. Two solutions were mixed and stirred with a magnetic bar until the homogeneous solution was obtained. 1-Methylimidazole was added into each solution for quaternization with chloromethyl groups with a PECH backbone, which would remain after the crosslinking reaction. The reaction proceeded at 100 °C overnight after the solution was poured on glass petri dishes. After crosslinking, the products were dried at 60 °C for 12 h in atmosphere, and then at 85 °C for 24 h under vacuum. After the membranes were peeled off from the petri dishes, they were immersed in 1 M NaOH aqueous solution for alkalization for 24 h. The membranes were then washed with distilled water a few times and then immersed in water for storage. The resulting membranes were homogeneous and transparent. Scheme 2 shows the synthesis procedure of the QPECH/PAEK-API. In this study, three types of QPECH/PAEK-APIs were synthesized with different molar ratios: 5.8 (PECH/PAEK-API1), 3.8 (QPECH/PAEK-API2) and 2.9 (QPECH/PAEK-API3).

### 2.3. Characterization

#### 2.3.1. Chemical Structure Analysis

^1^H nuclear magnetic resonance (^1^H NMR, Varian Unity INOVA 500 MHz, Varian, Palo Alto, CA, USA) spectrometer was employed to analyze the chemical structure of PAEK, PAEK-NHS and PAEK-API. For this measurement, the samples were dissolved in DMSO-d_6_ solvent with a tetramethylsilane internal standard. Gel permeation chromatography (GPC, Agilent 1100S, Santa Clara, CA, USA) was employed to measure the number and weight average molecular weights of PAEK-API. In this measurement, the solvent was THF, and the feed flow rate of injected solution was 1 µL/min^−1^.

#### 2.3.2. Ion Exchange Capacity (IEC)

The membrane samples in –OH form were washed with water and dried to measure weight, W_d_ (in gram), before being immersed in a HCl solution under an N_2_ gas environment for ionic exchange to Cl form. The back titration was conducted using NaOH solution with phenolphthalein indicator. The IEC (mequiv g^−1^) values of membranes were calculated from Equation (1):(1)$$\mathrm{EC}=\frac{M_{1}V_{1}{-M}_{2}V_{2}}{W_{d}}$$

Here, M_1_ (mol mL^−1^) and V_1_ (mL) are the molar concentration and volume of the HCl solution, and M_2_ and V_2_ are those of the NaOH solution, respectively.

#### 2.3.3. Anion Conductivity

The membranes were immersed in water and then cut into 3 cm (length) × 1 cm (width) × ~100 μm (thickness) dimension to measure anion conductivity. The sample was placed in the 4-probe cell (BEKKTECH, USA) and the in-plane anion conductivity was measured by alternating current (AC) impedance spectroscopy (Zahner IM6e, Germany) at the frequency range from 1 Hz to 1 MHz, at 5 mV, under 100 % relative humidity. The bulk resistance of the membrane was directly obtained from the impedance curve, and the hydroxide ion conductivity of the membrane was determined from the resistance using Equation (2):(2)$$\sigma =\frac{L}{Z\;W\;T}$$

Here, σ is the hydroxide ion conductivity of the membrane in S/cm, L is the distance in the direction of the ion flow between the measurement probes in cm, Z is the bulk resistance of the membrane in ohm, W is the width of the membrane in cm, and T is the thickness of the membrane in cm.

#### 2.3.4. Water Uptake and Length Swelling Percentage

The membrane sample (3 cm × 1 cm) was completely dried at 80 °C for 24 h in a vacuum oven to measure its dry weight, W_d_. Then, the dry membrane sample was soaked in DI water until equilibrium uptake at different temperatures. When the sample weight was invariant, the membrane sample was removed from the water and its surface was quickly wiped with tissue to measure its wet weight W_s_. The water uptake was calculated from Equation (3):(3)$$\%\;\mathrm{Water}\;\mathrm{uptake}=\frac{W_{s}{-W}_{d}}{W_{d}}\times 100$$

Furthermore, the length swelling percentage was calculated from Equation (4):(4)$$\mathrm{Length}\;\mathrm{swelling}\;\mathrm{percentage}=\frac{L_{s}{-L}_{d}}{L_{d}}$$
where L_s_ is the length of the wet sample and L_d_ is the length of the dry sample.

#### 2.3.5. Ion Cluster Dimension

The average dimension of the ionic clusters, d, in the membranes was investigated using small angle X-ray scattering (SAXS) spectroscopy (SAXSess, Anton Paar GmbH, Austria). The membrane of the dimension 20 mm × 3 mm was immersed in DI water at room temperature. The membrane sample was placed inside the SAXS instrument with X-ray synchrotron radiation (λ = 0.154 nm). The operation was running for 30 min while the imaging plate was recording the signal of the exposing X-ray. The reader (Cyclone Plus, Perkin Elmer, Waltham, MA, USA) converted the image of the plate into digital data. The average ionic cluster dimension, d, was calculated from Equation (5):(5)$$d=\frac{2\pi }{q}$$
where q (nm) is the scattering vector.

#### 2.3.6. Thermal, Mechanical, and Chemical Stability

A thermogravimetric analyzer (TGA, Seiko Exstar 6000, Japan) was used to investigate the thermal stability of the QPECH/PAEK-API membranes. The dry membrane sample was thermally scanned at 10 °C/min^−1^ from 30 °C to 550 °C under a nitrogen atmosphere.

The samples were prepared into 4 cm × 1 cm dimension. The samples were soaked in water overnight at room temperature until equilibrium, before measuring the mechanical properties. After the sample surface was blotted dry with a soft tissue, the tensile property of the sample was measured by a universal tensile machine (UTM model 5565, Lloyd, Fareham, UK) with a load of 250 N.

Several pieces of QPECH/PAEK-API alkaline membranes (1 cm × 3 cm) were immersed in 4 M NaOH solution for a month at 60 °C. Each sample was taken out of the solution to be washed with water every week. After drying, the IEC value was measured by the method mentioned above. Four IEC values for each QPECH/PAEK-API membrane over a month were collected for their chemical stability analysis in alkaline solution.

## 3. Results and Discussion

### 3.1. Chemical Structure Analysis

PAEK-API was synthesized before crosslinking with PECH. The weight average molecular weight of PAEK-COOH was 6.68 × 10^4^ g/mol from GPC analysis. PAEK-NHS, an intermediate product, was produced by modification of PAEK-COOH for feasible conversion to PAEK-API. One part of the chloromethyl groups of PECH played a role in the crosslinking reaction with the imidazolium groups of PAEK-API, while the remaining chloromethyl groups of PECH were reacted with 1-methylimidazole for quarternization. ^1^H NMR spectra of the three samples of PAEK-COOH, PAEK-NHS and PAEK-API are shown in Figure 1. In Figure 1, the protons of the benzene rings of the PAEK backbone led to NMR signals at 7.05, 7.26, and 7.75 ppm. A characteristic proton signal from -COOH appeared at 12 ppm in the PAEK-COOH spectrum (Figure 1a), but it completely disappeared in the PAEK-NHS spectrum (Figure 1b). This result confirmed that all -COOH pendant groups in PAEK-COOH were entirely converted to –NHS groups, with the appearance of a new signal at 2.77 ppm in Figure 1b. In Figure 1c, the NMR spectrum of PAEK-API did not show any NHS signal at 2.77 ppm. Instead, there were new proton signals at 7.79, 7.56, 6.83, 4.07, 2.96 and 2.33 ppm from the API group.

The QPECH/PAEK-API cross-linked membrane was totally different in physical appearance from the QPECH and PAEK-API blend membrane, simply obtained by mixing PAEK-API and QPECH solutions. In Figure 2a,b, while the blend membrane is opaque, the cross-linked one is obviously transparent. This difference was caused by the compatibility difference between the PAEK-API and QPECH molecules in the blend and cross-linked structures. In blend system, the two polymer micro-phases were separated, but in the cross-linked system, the crosslinking via quaternized imidazole reaction created the homogeneous phase between them. The cross-linked structure was also confirmed by solubility. When the blend and cross-linked samples were separately immersed in DMAc solvent, the blend membrane was immediately dissolved, but the cross-linked one was insoluble in it. In this study, three types of QPECH/PAEK-API cross-linked membranes were synthesized, according to the molar ratio of QPECH to PAEK-API mentioned in Section 2.2.2.

### 3.2. Ion Exchange Capacity (IEC)

The back titration was carried out to measure the IEC values of three QPECH/PAEK-API membrane samples, and the results are shown in Table 1. As the crosslinking density increases (more PAEK-API molecules are involved in crosslinking), the fraction of free imidazolium groups decreases, and thus the IEC of the membranes decreases. From QPECH/PAEK-API1 to QPECH/PAEK-API3, the IEC values decrease from 1.25 to 0.78 mequiv g^−1^.

### 3.3. Water Uptake and Length Swelling Percentage

The water uptake and length swelling percentage of the QPECH/PAEK-API membranes at different temperatures are shown in Figure 3a,b, respectively. The water uptake and length swelling percentage of the QPECH/PAEK-API membranes increased with the increasing number of free imidazolium groups in QPECH, because it decreased the crosslinking density, but increased the hydrophilicity of the membrane. As the molar ratio of QPECH to PAEK-API increased from 2.9 to 5.8, the water uptake and length swelling percentage increased, from 21.2% to 29.8% and from 7.34% to 9.38% at 30 °C, and from 24.9% to 39.8% and from 9.17% to 12.5% at 80 °C, respectively.

### 3.4. Anion Conductivity

The ionic conductivity of AEMs is significantly affected by water uptake, as ions transport through water channels. The hydroxide ion conductivity of QPECH/PAEK-API membranes was tested using AC impedance spectroscopy. As this was performed in the frequency range of 1 Hz to 1 MHz, the corresponding frequency of the conductivity shown in Figure 4 reached the maximum frequency of 1 MHz.

As the API group of PAEK-API was partially cross-linked the chloromethyl group of PECH, the crosslinking density affected the conductivity of QPECH/PAEK-API membranes, because the increase in cross-linking density reduces the molecular mobility of the polymer chains. Figure 4 shows that the increasing number of free imidazolium groups (or decreasing crosslinking density) leads to increase in anion conductivity. The conductivity increased from 0.0045 S/cm to 0.022 S/cm at 30 °C, and from 0.0112 S/cm to 0.033 S/cm at 80 °C, as the molar ratio of QPECH to PAEK-API increased from 2.9 to 5.8. When the temperature increased (from 30 °C to 80 °C), the free volume and polymer molecular mobility increased, and thus the conductivity increased. The hydroxide ion conductivity of an AHA commercial membrane (Astom Corporation, Japan) was also tested for comparison. The anion conductivity of the QPECH/PAEK-API2 and QPECH/PAEK-API1 membranes was higher than that of the AHA membrane.

The activation energy of this conductivity was calculated from the linearized plot of the ln σ vs. 1/T, and the resulting values are summarized in Table 1. The activation energy of the QPECH/PAEK-API1 membrane is lower than that of the QPECH/PAEK-API3, because the presence of more imidazolium groups provides easier ion transportation. Simultaneously, the higher crosslinking in QPECH/PAEK-API3 increases the activation energy in association with the lower molecular mobility.

On the other hand, the corresponding real and imaginary parts of the Nyquist plots for the calculation of the bulk resistance are shown in Table 2. The bulk resistance was calculated from $Z=\sqrt{{\left(Z_{\mathrm{re}}\right)}^{2}+{\left(Z_{\mathrm{im}}\right)}^{2}}$.

### 3.5. Mechanical Property

Figure 5 shows the stress vs. strain behavior for the hydrated QPECH/PAEK-API membranes. The tensile strength of the QPECH/PAEK-API1 membrane, 7.37 MPa, was the lowest, while that of QPECH/PAEK-API3 membrane, 15 MPa, was the highest. The elongation at break, however, was the reverse (27.8% for QPECH/PAEK-API1 membrane, 13.7% for QPECH/PAEK-API3). These mechanical properties of the hydrated membrane were strongly related with the water uptake shown in Figure 3. As the concentration of PAEK-API increases, the crosslinking density increases too, but the number of free anion-conducting groups decreases. Thus, this leads to a decrease of water uptake, but an increase of tensile strength.

### 3.6. Thermal Property

The thermal degradation behaviors of the three QPECH/PAEK-API membranes are shown in Figure 6. The slight weight loss up to 250 °C was due to the evaporation of water bound to polymer molecules even after drying. The QPECH and PAEK-API backbones started to degrade at 380 °C, while the imidazolium groups did so at 250 °C. The weight loss behaviors of the membranes were almost invariant with the composition of QPECH/PAEK-API. All of the synthesized QPECH/PAEK-API membranes were thermally stable up to 250 °C, which is much higher than the practical operation temperature of fuel cells.

### 3.7. Ion Cluster Structure

Under full water uptake conditions, the imidazolium groups would form ionic clusters in the membrane structure. QPECH/PAEK-API membranes were immersed in water before Small-angle X-ray scattering (SAXS) measurement. The SAXS patterns of the hydrated QPECH/PAEK-API membranes are shown in Figure 7, where the strongest characteristic peak is displayed at the scattering vector (q) of 1.24 nm^−1^ (corresponding to the ionic cluster dimension of 4.91 nm). As the molar ratio of QPECH to PAEK-API decreased from 5.8 to 2.9, the scattering peaks became broader. This was because the membrane loses hydrophilicity with the decreasing density of free imidazolium groups. The PAEK-COOH membrane does not show a clear SAXs peak because of the absence of the API groups. This SAXS patterns are well correlated with the anion conductivity, mechanical properties and water uptake of the membranes aforementioned.

### 3.8. Chemical Stability

In order to examine the chemical stability of QPECH/PAEK-API membranes, the IEC values of membranes were periodically measured after the membranes were placed in the basic pH environment for a month. In Figure 8, as the IECs of those membranes were almost invariant, we confirmed that the QPECH/PAEK-API membranes synthesized in this work displayed a good chemical stability.

## 4. Conclusions

In this study, PAEK-API was cross-linked with PECH to prepare a QPECH/PAEK-API alkaline exchange membrane, with both highly conductive and mechanically robust properties. This work investigated the effect of the QPECH to PAEK-API molar ratio on membrane properties, such as anion conductivity, water uptake, length swelling percentage and tensile strength, as the crosslink density increases but the number of free imidazolium group decreases with it. A higher conductivity and water uptake, but lower tensile strength, was observed at the lower QPECH/PAEK-API molar ratios. The free imidazolium groups in the hydrated membranes would create ionic clusters with various dimensions, as shown by SAXS analysis, and this directly affects the membrane properties. As QPECH/PAEK-API1 and QPECH/PAEK-API2 membranes showed quite well-balanced properties overall, these are expected to be applied in anion exchange fuel cells.

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
