# Peer review of "Anion Exchange Membranes Prepared from Quaternized Polyepichlorohydrin Cross-Linked with 1-(3-aminopropyl)imidazole Grafted Poly(arylene ether ketone) for Enhancement of Toughness and Conductivity"

_membranes, 2020, doi:10.3390/membranes10070138_

Round 1

Reviewer 1 Report

The manuscript reports a novel anion exchange membrane from quaternized polyepichlorohydrin cross-linked 1-(3-aminopropyl)imidazole-grafted-poly(arylene ether ketone). The work is of interest. I suggest its acceptance and publication in the Membranes, if the following concerns have been addressed.

(1) Figure 4b should be presented in form of point-line diagram.

(2) The tables of some properties, such as IEC, water uptake, swelling ratio and mechanical performance should be added.

(3) The title claims “…… PEK for enhancement of toughness and conductivity”, but the reasonable reference membranes (e.g. the samples without PEK) are necessary to indicate the enhancement. And the conductivity of commercial membrane AHA is so low that this contrast is not rational.

(4) why does author choose PECH as matrix materials? The ether bonds are sensitive in alkaline condition.

(5) A further discussion about alkaline stability should be given, because PECH and 1-(3-aminopropyl) imidazole grafted poly(arylene ether ketone) have sensitive ether bond and amide bonds. It is unexpected that the cross-linked membranes show such excellent alkaline stability in 4 M NaOH at 60 oC.

(6) The experiment section should be improved, and the adding amounts of 1-methylimidazole and other compounds should be presented.

Author Response

Dear Reviewer

Thank you very much for your comment to our manuscript which submitted to Membranes (Manuscript ID Membranes-820061). As we have revised the manuscript according to reviewers’ comments, please find our revised manuscript submitted through the MDPI online submission site (File name Membranes-820061). The changes made in the manuscript have been highlighted in yellow for your convenience. Our reply was added after the reviewer’s comments

Manuscript ID: Membranes-820061

MS Type: Full Paper

Title: Anion exchange membranes prepared from quaternized polyepichlorohydrin cross-linked with 1-(3-aminopropyl) imidazole grafted poly(arylene ether ketone) for enhancement of toughness and conductivity

Correspondence Author: Prof. Dukjoon Kim

Reviewer 1:

The manuscript reports a novel anion exchange membrane from quaternized polyepichlorohydrin cross-linked 1-(3-aminopropyl)imidazole-grafted-poly(arylene ether ketone). The work is of interest. I suggest its acceptance and publication in the Membranes, if the following concerns have been addressed.

  • Figure 3b should be presented in form of point-line diagram.

Ans: The format of Figure 3b was changed to point-line diagram. (Section3.3, page 16)

  • The tables of some properties, such as IEC, water uptake, swelling ratio and mechanical performance should be added.

Ans:  The table of some properties (IEC, water uptake, swelling ratio and mechanical property) was added in the result and discussion section. (Table 1, page 15)

  • The title claims “…… PEK for enhancement of toughness and conductivity”, but the reasonable reference membranes (e.g. the samples without PEK) are necessary to indicate the enhancement. And the conductivity of commercial membrane AHA is so low that this contrast is not rational.

Ans:

-  Thanks for reviewer’s kind comment. For the same reason as the reviewer pointed out, we actually tried to synthesize and characterize the pristine PECH membrane as a reference membrane. As the pristine PECH is, however, mechanically too weak and too sensitive with the moisture, we could not prepare the PECH membrane (without PAEK) as a free standing solid film for further characterization. It had no value as a membrane.

- As we know, the proton conductivity of membrane is influenced by experimental condition such as type of instrument, measurement method (through and longitudinal), humidity, etc. Actually, the ion conductivity of AHA commercial membrane measured in this study was not quite different from those from some previous measurements (See the following references).

- Zhang, Chengxu, et al. "Microphase separated hydroxide exchange membrane synthesis by a novel plasma copolymerization approach." Journal of Power Sources 198 (2012): 112-116.

- Zhang, Chengxu, et al. "Pulsed plasma-polymerized alkaline anion-exchange membranes for potential application in direct alcohol fuel cells." Journal of Power Sources 196.13 (2011): 5386-5393.

  • why does author choose PECH as matrix materials? The ether bonds are sensitive in alkaline condition.

Ans: As presented in Introduction, PECH is an easily purchasable commercial polymer with cost competiveness and some prominent properties such as good thermal stability and proton conductivity. Especially, the highly concentrated pendant chloromethyl groups in PECH are quite attractive in establishing high anion conductivity by a simple quaternization reaction without external chloromethylation. This is the main reason why we chose it as a major component. As the mechanical strength of the pristine PECH is too low, it was cross-linked with PAEK-API. Some references are added here regarding application of PECH.

- Yang, Chun-Chen. "Alkaline direct methanol fuel cell based on a novel anion-exchange composite polymer membrane." Journal of Applied Electrochemistry 42.5 (2012): 305-317.

- Yang, Chun-Chen, Sheng-Jen Lin, and Sung-Ting Hsu. "Synthesis and characterization of alkaline polyvinyl alcohol and poly (epichlorohydrin) blend polymer electrolytes and performance in electrochemical cells." Journal of power sources 122.2 (2003): 210-218.

- Yu, E. H., and K. Scott. "Direct methanol alkaline fuel cells with catalysed anion exchange membrane electrodes." Journal of applied electrochemistry 35.1 (2005): 91-96.

  • A further discussion about alkaline stability should be given, because PECH and 1-(3-aminopropyl) imidazole grafted poly(arylene ether ketone) have sensitive ether bond and amide bonds. It is unexpected that the cross-linked membranes show such excellent alkaline stability in 4 M NaOH at 60 o

Ans: As we also worried about the chemical stability of amide bond, we have examined it conducting many experiments. The following studies have revealed that the ether and amide bonds in this type of PAEK graft system still have good chemical stability in alkaline condition.

- Lee, Nakwon, Diem Tham Duong, and Dukjoon Kim. "Cyclic ammonium grafted poly (arylene ether ketone) hydroxide ion exchange membranes for alkaline water electrolysis with high chemical stability and cell efficiency." Electrochimica Acta 271 (2018): 150-157.

- Tuan, Cao Manh, and Dukjoon Kim. "Anion-exchange membranes based on poly (arylene ether ketone) with pendant quaternary ammonium groups for alkaline fuel cell application." Journal of membrane science 511 (2016): 143-150.

- Tuan, Cao Manh, Astam K. Patra, and Dukjoon Kim. "Chemically modified poly (arylene ether ketone) s with pendant imidazolium groups: Anion exchange membranes for alkaline fuel cells." International Journal of Hydrogen Energy 43.9 (2018): 4517-4527.]

Lee et al. pointed out that the IEC values of PAEK grafted 1-(3-aminopropyl)-4-methylpiperazine slightly decreased during 1 week in the NaOH (10%) solution at 60oC but kept constant in the following 3 weeks. Moreover, The FT-IR spectra of PAEK membranes before and after chemical stability test were quite similar each other. The IR band of the grafted functional groups at around 1150 cm-1 after 4 weeks in the highly concentration alkaline concentration was still similar to that of the original one.

  • The experiment section should be improved, and the adding amounts of 1-methylimidazole and other compounds should be presented.

Ans: The amounts of 1-methyllimidazole and other compounds were added in the experimental section. (Section 2.2, page 5)

Reviewer 2 Report

In the manuscript titled “Anion exchange membranes prepared from quaternized polyepichlorohydrin cross-linked with 1-(3-aminopropyl) imidazole grafted poly(arylene ether ketone) for enhancement of toughness and conductivity”, the authors synthesized an anion-exchange membrane by cross-linking QPECH and PAEK-API. They conducted series studies of different properties of the anion-exchange membrane made of the new polymer. Their approach is interesting. The designs and experiments are reasonable. The writings of the manuscript are also fine. I think this paper is publishable after some changes, as listed below.

  • In line 134, the thickness is 100cm. This seems to be quite large. Please verify.
  • In line 148, how did they assure that the uptake equilibrated? Please specify the method they used to measure the weight change of the membrane immersed in water.
  • In eqn (4), why only the change of length of the membrane is taken into account? Are there changes in width and thickness? Usually, people use volume to define swelling ratio.
  • In the measurement of IEC, they conducted experiments in HCl solution. Since the protonation state of the imidazolium may influence its ability of anion adsorption, is it necessary to study the capability of anion adsorption at other pH?
  • In characterizing the Anion conductivity, they used AC with different frequency. But they only gave one value of conductivity. What’s the corresponding frequency in Fig. 5? In addition, it would be useful to give the full impedance spectroscopy results. It will help to elucidate the nature of anion transport in the membrane.
  • Also for conductivity characterization. As they conducted experiments at different temperature in Figure 5, it’s useful to extract some transport parameters, such as activation energy of transport by fitting the data. This will also provide in-depth understanding of the ion transport mechanisms.
  • In line 235-240, they explained the increase in conductivity with decreasing cross-linking by the amount the free imidazolium groups. There is another factor, which is the chain flexibility. Will chain flexibility also influence?
  • The Ion cluster structure section seems problematic. The peak is not very obvious. Even if there is peaks, how did they justify that the peak is from ion cluster rather than polymer packing?

Author Response

Dear Reviewer

Thank you very much for your comment to our manuscript which submitted to Membranes (Manuscript ID Membranes-820061). As we have revised the manuscript according to reviewers’ comments, please find our revised manuscript submitted through the MDPI online submission site (File name Membranes-820061). The changes made in the manuscript have been highlighted in yellow for your convenience. Our reply was added after the reviewer’s comments

Manuscript ID: Membranes-820061

MS Type: Full Paper

Title: Anion exchange membranes prepared from quaternized polyepichlorohydrin cross-linked with 1-(3-aminopropyl) imidazole grafted poly(arylene ether ketone) for enhancement of toughness and conductivity

Correspondence Author: Prof. Dukjoon Kim

Reviewer 2:

In the manuscript titled “Anion exchange membranes prepared from quaternized polyepichlorohydrin cross-linked with 1-(3-aminopropyl) imidazole grafted poly(arylene ether ketone) for enhancement of toughness and conductivity”, the authors synthesized an anion-exchange membrane by cross-linking QPECH and PAEK-API. They conducted series studies of different properties of the anion-exchange membrane made of the new polymer. Their approach is interesting. The designs and experiments are reasonable. The writings of the manuscript are also fine. I think this paper is publishable after some changes, as listed below.

(1) In line 134, the thickness is 100cm. This seems to be quite large. Please verify.

Ans: Thanks for the comment. It was a typo error. It was changed from cm to . (Section 2.3.3, page 9)

(2) In line 148, how did they assure that the uptake equilibrated? Please specify the method they used to measure the weight change of the membrane immersed in water.

Ans: If the water uptake of the membrane reaches equilibrium, the weight of membrane is the constant value. In the water uptake test, the dry membrane was immersed in water, and its weight was periodically measured until no weight change was observed. As Ws was measured by this method, we are sure that the water uptake is equilibrium value.

(3) In eqn (4), why only the change of length of the membrane is taken into account? Are there changes in width and thickness? Usually, people use volume to define swelling ratio.

Ans:

- Because the length and width of the membrane (3 cm x 1 cm) was much larger than the thickness of 100 m, the difference in length dimension before and after swelling was much more apparent that that in thickness. That is the reason why the authors focused its length for swelling measurement.

-  As the thickness of membrane was very thin , its swelling (thickness increase)  was too small for precise measurement, This is the reason the authors define the swelling ratio with the length in this manuscript.

 (4) In the measurement of IEC, they conducted experiments in HCl solution. Since the protonation state of the imidazolium may influence its ability of anion adsorption, is it necessary to study the capability of anion adsorption at other pH?

Ans: In the IEC measurement, 0.3 g dry membrane was immersed in 20 mL of 0.01 N HCl for 48 h. After that, the membrane was removed from this solution and then titrated with 0.01 N NaOH standard solution using a pH meter. Because of this measurement process, the protonation state of imidazolium from HCl solution could not influence the IEC result.

(5) In characterizing the Anion conductivity, they used AC with different frequency. But they only gave one value of conductivity. What’s the corresponding frequency in Fig. 5? In addition, it would be useful to give the full impedance spectroscopy results. It will help to elucidate the nature of anion transport in the membrane.

Ans:

- In this study, the conductivity measurements were performed in the frequency range from 1 Hz -1 MHz at 5mV as already mentioned in the experiment section. (Section 2.3.3, page 9). Thus, the corresponding frequency of the conductivity in Fig. 4 is the maximum frequency of 1 MHz.

- As the computer software provides the bulk resistance calculated from from the real and imaginary parts of Nyquist plots, we did not download all data at different frequencies. However, as we still have the real and imaginary parts for the resulting bulk resistance, we added those in Table 2 (Section 3.4, page 18).

(6) Also for conductivity characterization. As they conducted experiments at different temperature in Figure 4, it’s useful to extract some transport parameters, such as activation energy of transport by fitting the data. This will also provide in-depth understanding of the ion transport mechanisms.

Ans: The activation energy was calculated and the result was added in Table 1. (Page 15)

(7) In line 235-240, they explained the increase in conductivity with decreasing cross-linking by the amount the free imidazolium groups. There is another factor, which is the chain flexibility. Will chain flexibility also influence?

Ans: In general, the degree of cross-linking of the polymer significantly influences the chain flexibility. Thus, both degree of cross-linking and chain flexibility have similar effect on the proton conductivity of membranes. We added molecular mobility of polymer chain as another factor influencing the conductivity. (Page 16-17)

(8) The Ion cluster structure section seems problematic. The peak is not very obvious. Even if there is peaks, how did they justify that the peak is from ion cluster rather than polymer packing?

Ans: The SAXs pattern of PAEK-COOH sample (without imidazole groups) was measured and added in Figure 7. The SAXs peaks were all from the ionic clusters formed by API groups rather than polymer packing, as the polymer membrane without functional groups did not show any SAXs peak.  (Page 21)

Round 2

Reviewer 1 Report

The manuscript improved a lot,and should be accepted.